# From Metabolic to Epigenetic Memory: The Impact of Hyperglycemia-Induced Epigenetic Signature on Kidney Disease Progression and Complications

**DOI:** 10.3390/genes16121442

**Published:** 2025-12-02

**Authors:** Sara Cannito, Ida Giardino, Maria D’Apolito, Alessandra Ranaldi, Francesca Scaltrito, Massimo Pettoello-Mantovani, Annamaria Piscazzi

**Affiliations:** 1Laboratory Medicine, Department of Clinical and Experimental Medicine, University of Foggia, 71122 Foggia, Italy; ida.giardino@unifg.it (I.G.); annamaria.piscazzi@unifg.it (A.P.); 2Medical Genetics, Department of Clinical and Experimental Medicine, University of Foggia, 71122 Foggia, Italyalessandra.ranaldi@unifg.it (A.R.); 3Residency Course in Pediatrics, Department of Medical and Surgical Sciences, University of Foggia, 71122 Foggia, Italy; 4Italian Academy of Pediatrics, 20126 Milan, Italy; 5Department of Pediatrics, Institute for Scientific Research «Casa Sollievo», University of Foggia, 71122 Foggia, Italy; 6European Pediatric Association, Union of National European Pediatric Societies and Associations, 10115 Berlin, Germany

**Keywords:** chronic kidney disease, cardiovascular complications, oxidative stress, mitochondrial alterations, metabolic memory, epigenetics, epigenetic modifications, epigenetic memory

## Abstract

Chronic kidney disease is a significant global health burden and a leading cause of cardiovascular morbidity and mortality. Diabetes mellitus is the primary cause of kidney disease, driving the progression of both micro- and macrovascular complications. Sustained hyperglycemia initiates a cascade of deleterious molecular and cellular events, including mitochondrial dysfunction, inflammation, oxidative stress, and dysregulated apoptosis and autophagy, which collectively contribute to the progression of renal injury. Beyond these well-established mechanisms, a compelling body of evidence highlights the pivotal role of epigenetic alterations (such as DNA methylation, histone post-translational modifications, and non-coding RNAs) in mediated long-term kidney damage. The interplay between transcriptional and epigenetic regulation underlies the phenomenon of the “metabolic memory”, wherein cellular dysfunction persists even after glycemic control is achieved. This review synthesizes the current knowledge on mechanisms sustaining metabolic and epigenetic memory, with a particular focus on the epigenetic machinery that establishes and maintains these signals, a concept increasingly termed “epigenetic memory.” Given their reversible nature, epigenetic determinants are emerging as promising biomarkers and a compelling therapeutic avenue. Targeting these “epifactors” offers a novel strategy to halt progression to end-stage renal disease, thereby paving the way for precision medicine approaches in diabetes-related renal disease.

## 1. Introduction

Kidney diseases, encompassing both acute kidney injury (AKI) and chronic kidney disease (CKD), represent a major global health burden and recognized as leading contributors to cardiovascular morbidity and mortality [1,2]. CKD is frequently associated with a higher incidence of comorbid conditions, such as hypertension, diabetes mellitus, glomerulonephritis, chronic pyelonephritis, autoimmune diseases, and recurrent urinary tract infections, whose progression is associated with the development of renal impairment and eventual kidney failure [1,3]. Globally, in 2017, the Global Burden Disease, Injuries, and Risk Factors (GBD) study showed that impaired fasting plasma glucose and high glucose pressure account for 57.6% and 43.2% of the age-standardized rate of CKD, respectively [4]. Both AKI and CKD arise from complex systemic interactions and cellular disturbances, involving hemodynamic alterations, inflammatory responses, and maladaptive repair mechanisms that perpetuate the renal injury. CKD progression, in particular, is driven by endothelial dysfunction, vascular rarefaction, inflammation, and progressive nephron loss [5]. Recent guideline frameworks have reinterpreted CKD as a highly prevalent condition with a continuum of severity, no longer viewed solely as a specialist disease but as a public health priority requiring early detection, preventive strategies, and multidisciplinary management [6]. According to the Kidney Disease Outcomes Quality Initiative (KDOQI) and Kidney Disease: Improving Global Outcomes (KDIGO) guidelines, CKD is defined by the presence of kidney damage or a reduction in glomerular filtration rate (GFR) for at least three months, regardless of etiology. Decreased kidney function is typically assessed through estimated GFR (eGFR), derived from serum creatinine–based equations. CKD often progresses silently, with patients remaining asymptomatic until substantial functional decline occurs. In a subset of individuals, disease progression culminates in end-stage renal disease (ESRD), characterized by severely reduced kidney function or dependence on dialysis. Despite advances in therapeutic strategies, the coexistence of diabetes remains a critical determinant of adverse cardiovascular and renal outcomes. Diabetic kidney disease (DKD) is a major microvascular complication of both type 1 and type 2 diabetes and a leading cause of ESRD worldwide [7]. DKD is characterized by a constellation of structural and functional changes, including podocyte effacement, mesangial expansion, excessive deposition of extracellular matrix (ECM), and tubular epithelial-to-mesenchymal transition (EMT) [8]. Alongside hypertension, sustained high glucose levels are the principal pathogenic driver of DKD [9]. Chronic hyperglycemia, orchestrates a complex and multifactorial cascade involving mitochondrial dysfunction, oxidative stress, inflammation, and dysregulated apoptosis and autophagy—processes that collectively promote cellular injury and fibrosis [10,11,12]. While the classical biochemical pathways underlying diabetic nephropathy have been extensively characterized, emerging evidence highlights the pivotal role of epigenetic mechanisms—including DNA methylation, histone post-translational modifications, and non-coding RNAs—in initiating and perpetuating renal dysfunction and its complications [13]. Epigenetic dysregulation affects multiple renal cell types, including mesangial cells, podocytes, tubular epithelial cells, and glomerular endothelial cells [14]. These alterations, which do not change the DNA sequence itself, modify gene expression patterns, leading to cellular damage and fibrosis, typical hallmarks of DKD. These environmentally responsive mechanisms may mediate the sustained expression of genes and phenotypes associated with DKD. These alterations, which do not modify the DNA sequence itself, alter gene expression patterns, driving cellular damage and fibrotic remodeling—hallmarks of DKD. Importantly, these environmentally responsive mechanisms may sustain the expression of pathogenic genes and phenotypes associated with DKD even after restoration of normoglycemia. The interplay between transcriptional and epigenetic factors underlies the phenomenon known as “metabolic memory,” a term coined by Michael Brownlee to describe the long-term deleterious effects of prior hyperglycemic exposure that persist despite subsequent glycemic control [15].

This review aims to summarize the current understanding of the molecular bases underlining the metabolic memory, emphasizing on epigenetic machinery mechanisms responsible for its establishment and maintenance—collectively referred to as “epigenetic memory”. Given the reversible nature of epigenetic modifications, “epifactors” hold promise as both biomarkers and therapeutic targets for hyperglycemia-related CKD. Modulating epigenetic regulators may open new avenues for halting disease progression, preventing ESRD, and advancing precision medicine approaches in DKD.

## 2. The Metabolic Memory

There is a complex interplay of molecular pathways activated by chronic hyperglycemia during the progression of diabetic kidney disease (DKD). Sustained exposure to elevated glucose levels triggers a cascade of intricate and deleterious molecular and cellular events that ultimately culminate in widespread cellular and tissue injury. The detrimental effects of glucotoxicity are mediated through multiple, interconnected mechanisms, including the accumulation of advanced glycation end products (AGEs), excessive generation of reactive oxygen species (ROS), and profound bioenergetic dysfunctions. These processes ultimately lead to epigenetic remodeling and the establishment of inflammatory memory [16] (Figure 1).

Collectively, these ROS-associated pathways contribute to glomerular cell dysfunction, renal inflammation, and fibrosis by promoting DNA damage, mitochondrial impairment, lipid peroxidation, and abnormal protein modifications [17].

Chronic hyperglycemia accelerates the formation of AGEs, which are closely linked to cardiovascular and microvascular complications [15,18]. The AGEs binding to their receptor (RAGE) activates multiple intracellular signaling pathways—including MAPK, ERKs, JNK, NF-κB, TGF-α, and NOX-1—which amplify oxidative stress, inflammation, and apoptosis [18,19,20,21]. Interestingly, transient or prolonged hyperglycemic insults can leave a persistent molecular fingerprint that continues to drive diabetic complications, like DKD, even after the normoglycemia is restored [22,23,24,25,26]. The first experimental evidence supporting this phenomenon came in vitro studies demonstrating self-perpetuating changes in the gene expression induced by transient hyperglycemia. Human endothelial cells exposed to high glucose for 14 days, followed by a 7-day normoglycemic period, continued to overexpress fibronectin mRNA—an important mediator of diabetic retinopathy—for several weeks after glucose normalization [27]. These observations early mechanistic insight into what is also known as “legacy effect” or “glycemic memory”.

### 2.1. Clinical Evidence from Landmark Trials

The concept of metabolic memory is strongly supported by longitudinal follow-up data from two landmark clinical trials: the Diabetes Control and Complications Trial (DCCT) and the UK Prospective Diabetes Study (UKPDS). In the DCCT, patients with type 1 diabetes who received intensive glucose control exhibited a significant reduction in microvascular complications compared with those receiving conventional therapy. Remarkably, during the subsequent Epidemiology of Diabetes Interventions and Complications (EDIC) follow-up study, glycemic levels between the two groups converged. Crucially, however, the initial benefit of intensive therapy on reducing the risk of microvascular complications (retinopathy, nephropathy, and neuropathy), not only persisted, but often widened over the subsequent two decades [28]. A similar effect was observed in the follow-up of the UKPDS cohort of type 2 diabetic patients. The early intensive glucose management conferred a lasting reduction in myocardial infarction and all-cause mortality, even after glycemic differences dissipated. Together, these findings demonstrated that early and tight glycemic control establishes a durable protective imprint, underscoring the need for early intervention to prevent the establishment of this pathogenic memory [28].

### 2.2. Oxidative Stress and Mitochondrial Dysfunction

From an experimental standpoint, memory reflects a sustained vascular stress signature characterized by increased ROS production, endothelial dysfunction, endoplasmic reticulum stress, and persistent activation of pro-atherogenic pathways [29]. Extensive evidence links oxidative stress and diabetic complications across multiple tissues [15,30,31]. Among the various sources of ROS, mitochondrial-derived ROS (mtROS) play a particularly critical role in diabetic vascular and renal pathology [24,32,33,34]. mtROS influence endothelial function, nitric oxide bioavailability, and redox-sensitive signaling cascades [34,35]. Ceriello and colleagues demonstrated that metabolic memory originates at the level of glycated mitochondrial proteins, further involving non-enzymatic glycation of cellular proteins, lipids, and nucleic acids [22]. Consistent with these observations, inhibition of oxidative stress through antioxidant therapy has been shown to attenuate diabetic complications [15,30,32]. Normalization of mtROS levels alleviates key hyperglycemia-induced pathways, including the polyol pathway, protein kinase C (PKC) activation, and AGEs accumulation [32]. Consequently, reducing mtROS production can “erase” the hyperglycemia-induced metabolic imprint, confirming the pivotal role of mitochondrial dysfunction in metabolic memory [23].

Glycation of mitochondrial components can trigger a deleterious cycle of mitochondrial DNA (mtDNA) damage and bioenergetic decline, perpetuating ROS overproduction and cellular injury [36]. Kowluru and colleagues reported that hyperglycemia alters mitochondrial dynamics, reducing the fusion protein Mitofusin 2 (Mfn2) and increasing the fission protein Dynamin-Related Protein 1 (Drp1), resulting in mitochondrial fragmentation. Restoration of normoglycemia failed to reverse these alterations, suggesting the persistence of dysfunctional mitochondria. Therapeutic modulation of mitochondrial fusion has been proposed as a means to restore mitochondrial quality control after high-glucose exposure [37].

Compromised mitochondrial integrity promotes cellular senescence and impairs mitophagy, as shown in retinal endothelial cells in diabetic retinopathy models [37,38,39]. Given the kidney’s high energy demand, similar mechanisms are pivotal in CKD. Chronic oxidative and nephrotoxic insults disrupt mitochondrial biogenesis, dynamics, and clearance, driving progressive renal dysfunction [40]. Moreover, mitochondrial impairment contributes to systemic metabolic disturbances, including inflammation, endothelial dysfunction, and insulin resistance [41,42,43]. Targeting mitochondrial quality control and metabolic memory regulators represents a promising therapeutic approach for limiting CKD progression and related complications [26,44,45]. Pharmacological agents such as mitochondrial division inhibitor 1 (Mdivi-1) and leflunomide, have shown efficacy in preserving mitochondrial structure and function in endothelial cells [37,46]. Notably, leflunomide administered during the hyperglycemia prevented mitochondrial damage, while post-hyperglycemia treatment restored mitochondrial dynamics, reactivated the mitophagy machinery, reduced ROS production, and inhibited apoptosis [37].

### 2.3. Impact of Glucose Fluctuations

An additional aspect of metabolic memory relates to glycemic variability. Intermittent glucose spikes can exert more deleterious effects than sustained hyperglycemia. Several studies have demonstrated that glucose oscillations elicit greater oxidative stress and endothelial dysfunction than sustained hyperglycemia, and that these detrimental effects persist even after glucose levels return to normal. Such intermittent glucose fluctuations promote a long-lasting pro-inflammatory and dysfunctional endothelial phenotype, driven by exacerbated oxidative stress and persistent epigenetic reprogramming [23,47]. Clinically, high glycemic variability is now recognized as an independent predictor of diabetes-related complications and mortality, beyond glycated haemoglobin (HbA1c) levels [48]. Experimental evidence indicates that transient hyperglycemia can induce epigenetic modifications of histone tails (e.g., methylation and acetylation) that persist after return to normoglycemia [49,50]. These stable epigenetic marks are increasingly recognized as key mediators of long-lasting detrimental effects of hyperglycemia in CKD pathophysiology and in the transition from AKI to CKD [51,52,53].

### 2.4. Therapeutic Implications

Understanding the mechanisms governing metabolic memory underscores the clinical importance of early, intensive glycemic control to rapidly restore metabolic homeostasis. Complementary strategies attenuating oxidative stress and inhibiting non-enzymatic glycation may further enhance long-term outcomes. Integrating such approach could form the foundation of more effective interventions aimed at preventing the long-term sequelae of hyperglycemia and limiting the progression of DKD.

## 3. Epigenetic Regulation of the Gene Expression

Understanding the hierarchical organization of DNA within chromatin provides the fundamental basis for elucidating the mechanisms of epigenetic regulation. Epigenetic mechanisms refer to potentially reversible and heritable modifications in chromatin structure and gene expression that drive tissue-specific transcription programs and maintain cellular identity, without altering the underlying DNA sequence [54]. Through coordinated biochemical modifications of DNA and histone or non-histone proteins, together with the action of chromatin remodelers and regulatory complexes, specific genes can be switched “on” or “off”, in response to environmental and cellular signals.

### 3.1. Chromatin Structure and DNA Methylation

Histone proteins constitute the core components of chromatin, whose fundamental unit is the nucleosome—147 base pairs of DNA wrapped around a histone octamer, consisting of two copies for each H2A, H2B, H3, and H4, while H1 serves as a linker histone (Figure 2A). In higher eukaryotes, principal epigenetic modifications include DNA methylation, post-translational modifications (PTMs), chromatin remodeling, histone variant exchange, and regulation by non-coding RNAs.

Among these, DNA methylation is the most extensively studied mechanism. It involves the covalent addition of a methyl group (-CH_3_) to the 5′ carbon of cytosine residues, generating 5-methylcytosines (5mC), primarily within CpG islands—short DNA regions (<500 bp) with high cytosine-guanosine content (>55%), commonly clustered around gene promoters (Figure 2B). This modification is catalyzed by DNA methyltransferases (DNMTs), which establish and maintain methylation patterns critical for normal tissues development and differentiation [55,56] (Figure 2B).

DNMTs are classified into two main functional groups: those responsible for de novo methylation (DNMT3A, DNMT3B) and those that maintain methylation during DNA replication (DNMT1). DNMT3L, although catalytically inactive, acts as a regulatory cofactor [57,58,59]. All of these enzymes catalyze the reaction through the s-adenosyl-l-methionine (SAM) as methyl groups donor. Interestingly, DNMT expression is tissue-specific; for instance, DNMT3A primarily regulates hematopoietic differentiation, whereas DNMT3B contributes to cartilage homeostasis and ossification [60]. Understanding the context-dependent functions of DNMTs is crucial for elucidating their role in disease pathogenesis and exploring therapeutic targeting strategies.

DNA methylation marks are reversible through passive demethylation (failure to maintain methylation during replication) or active demethylation mediated by Ten-Eleven Translocation (TET) enzyme family (TET1, TET2, and TET3). These enzymes iteratively oxidize 5mC to 5-hydroxymethylcytosine and further intermediates, which are then excised and removed via the base excision repair pathway, restoring the unmethylated state [61,62,63] (Figure 2B).

### 3.2. Histone Post-Translational Modifications

Histone PTMs represent another essential layer of epigenetic regulation, controlling chromatin accessibility and thereby influencing transcriptional activity. The amino-terminal tails of histones are subject to diverse PTMs, including acetylation, methylation, phosphorylation, ubiquitylation, SUMOylation, crotonylation, lactylation, and others [64,65,66,67]. These modifications can either promote chromatin relaxation and gene activation or induce compaction and transcriptional repression (Figure 2C). For example, acetylation of lysine residues neutralized their positive charge, reducing the electrostatic interaction between histones and DNA, thereby loosening chromatin and facilitating transcription. Conversely, removal of acetyl groups enhances chromatin condensation into transcriptionally inactive heterochromatin [68].

Histone methylation is another key modification that can either activate or repress gene expression, depending on the residue and degree of methylation. Lysine methylation is catalyzed by lysine methyltransferases (KMTs) and reversed by lysine demethylases (KDMs), which have been implicated in neurological disorders, inflammation, cancer, and metabolic disease [69,70,71]. Canonical methylation sites include H3K4, H3K9, H3K27, H3K36, H3K79, and H4K20, each carrying out distinct regulatory effects on chromatin structure and transcription [72].

### 3.3. Polycomb Repressive Complexes and Chromatin Remodeling

The stable maintenance of repressive chromatin states during development relies on recruitment of silencing complexes, such as methyl-CpG-binding proteins (e.g., MeCP2) or the Polycomb Repressive Complex 2 (PRC2) [73,74]. PRC2 catalyzes mono-, di-, and trimethylation of histone H3 at lysine 27 (H3K27me1/2/3), promoting chromatin compaction and transcriptional repression. Although the mechanisms governing PRC2 recruitment of specific genomic loci remain incompletely understood, its function is central to the propagation of silenced gene states [74,75].

Methylated histone residues are recognized by chromodomain-containing proteins, which translate histone marks into functional outcomes by assembling transcriptional activators or repressor [76]. For instance, the Polycomb complex is recognized by CBX chromodomain, leading to the monoubiquitination of histone H2A at lysine 119 (H2AK119ub) [77].

### 3.4. Histone Acetylation and Deacetylation

Histone acetylation is dynamically regulated by histone acetyl-transferases (HATs) and histone deacetylases (HDACs). HATs catalyze the transfer of an acetyl group (CH_3_CO–) from acetyl-CoA to the ε-amino group of lysine residues, a process recognized by bromodomain-containing proteins that facilitate transcriptional activation. Conversely, HDACs remove the acetyl groups, restoring positive charges on histones and promoting chromatin condensation and transcriptional repression [78].

Eighteen HDACs have been identified in humans, classified into four groups:Class I (HDAC1, 2, 3, 8);Class II (HDAC4, 5, 6, 7, 9, 10);Class III or Sirtuins (SIRT1-7);Class IV (HDAC11).

While classes I, II, and IV share structural and functional similarities, Sirtuins employ a unique NAD^+^-dependent deacetylation mechanism, producing nicotinamide and 2′-O-acetyl-ADP-ribose as metabolic products [78] (Table 1).

### 3.5. Interplay Between Epigenetic Mechanisms

Although mediated by distinct enzymatic machinery, DNA methylation and histone PTMs are functionally interconnected. Histone modifications have been shown to induce DNA methylation, a process especially observed during the early development also able to stabilize DNMTs [79,80]. Once established, methylated CpGs recruit methyl-CpG-binding proteins, such as MeCP2 and associated histone-modifying enzymes such as HDACs and KMTs, forming co-repressor complexes that reinforce gene silencing [79,81]. Moreover, the establishment of specific histone modifications can require rearrangements. For instance, before PRC2 can methylate H3K27, any pre-existing acetylation at this residue must be removed; this requires HDACs, which render the lysine’s group accessible for subsequent PRC2-mediated methylation, highlighting the dynamic interplay among these modifications [82].

### 3.6. Non-Coding RNAs and microRNA-Mediated Regulation

Non-coding RNAs, particularly microRNAs (miRNAs), have emerged as pivotal epigenetic regulators and biomarkers with significant diagnostic and prognostic potential across a variety of diseases [40,83]. For instance, several circulating miRNAs are dysregulated in the blood of cancer patients’, serving as potential biomarkers for early cancer diagnosis [40]. Given their established role in carcinogenesis, panels of miRNAs have been proposed as valuable tools for cancer screening and classification. For example, a specific five-miRNA signature—comprising miR-486-5p, miR-451, miR-92a, miR-25, and miR-16—has been identified for the early detection of gastric cancer [40].

Beyond their role in oncology, the prognostic and diagnostic relevance of miRNAs is increasingly recognized in a wide spectrum of other pathological conditions, including renal and metabolic diseases [84]. Altered expression profiles of circulating miRNAs are emerging as promising non-invasive biomarkers for predicting the progression of CKD and assessing the prognosis of metabolic disorders such as diabetes and its associated complications, including diabetic nephropathy and cardiovascular disease [85,86].

At the molecular level, miRNAs are small (18–25 nucleotide), endogenous, single-stranded RNA molecules that act as post-transcriptional regulators of gene expression. They originate from hairpin-loop precursors that undergo a series of tightly regulated processing steps. Most miRNAs are transcribed by RNA polymerase II as long primary transcripts (pri-miRNAs), which are cleaved in the nucleus by the Drosha–Pasha (DGCR8) complex to produce precursor miRNAs (pre-miRNAs). These pre-miRNAs are subsequently exported to the cytoplasm, where the RNase III enzyme Dicer further processes them into short miRNA duplexes. One strand of this duplex, known as the guide strand, is incorporated into the Argonaute-containing RNA-induced silencing complex (RISC). Through this complex, miRNAs exert their regulatory function by binding to complementary sequences in target mRNAs, leading to either mRNA degradation or translational repression, depending on the degree of sequence complementarity [87].

### 3.7. Epigenetics and Disease Implications

Aberrant epigenetic regulation contributes to multiple pathological conditions, including cancer, fibrosis, autoimmune disorders, inflammatory diseases, and metabolic disorders [88,89]. Persistent epigenetic marks can sustain maladaptive gene expression long after the initial trigger, contributing to the progressive nature of CKD and other metabolic complications. Importantly, the reversible nature of epigenetic modifications offers promising opportunities for therapeutic intervention and prognostic innovation, particularly in preventing or mitigating the metabolic memory phenomenon in diabetic and renal disease.

## 4. The Role of the Epigenetic Memory in Hyperglycemia-Related CKD Progression

Epigenetics serves as a crucial mediator between genetic predisposition and environmental or lifestyle factors—such as diet, exercise, oxidative stress, toxins, metabolic alterations, and inflammation—thereby influencing disease onset and progression. In the context of renal disease, oxidative stress, inflammation, and metabolic dysfunction are tightly linked to extensive epigenetic remodeling, which heightens the risk of developing ESRD and related complications. This association is particularly pronounced in the presence of hyperglycemia or diabetes. DKD represents a paradigmatic example in which sustained hyperglycemia induces profound epigenetic alterations—such as DNA methylation and histone modifications—that reprogram pathological gene expression [9,90]. Notably, histone modifications observed in the proximal tubular cells of diabetic mice persist even after normalization of glycemia, underscoring the long-lasting nature of these epigenetic changes [91,92].

The investigation of epigenetic processes contributing to metabolic memory of CKD progression and its cardiovascular complications is an emerging and rapidly expanding field. Multiple biochemical and signaling pathways are disrupted in CKD, including those governing inflammation, immune modulation, EMT, and metabolic homeostasis, key drivers of renal fibrosis and vasculopathy, including accelerated atherosclerosis and vascular calcification [93,94].

### 4.1. Histone Modifications and the Activation of Pro-Inflammatory Signaling

Epigenetic modulation involves alterations to DNA accessibility, determined in part by post-translational modifications of core histones. These histone modifications are intrinsically linked to the development and progression of DKD, contributing to localized oxidative stress, fibrosis, and inflammation within the renal environment [50]. Specifically, certain histone modifications can activate pro-inflammatory signaling pathways, thereby initiating sustained inflammation and structural damage within the kidney parenchyma [95]. In diabetic mouse models, reduced Suv39h1 levels led to decreased H3K9me3 deposition at inflammatory gene promoters, promoting gene derepression and persistent vascular inflammation [96]. Hyperglycemia enhances activating histone marks (H3K9ac, H3K4me1, and H3K4me3) while diminishing repressive marks (H3K27me3) at promoters such as TXNIP, a key regulator of oxidative stress and inflammation [97]. Preclinical evidence from DKD models also indicates that genes such as plasminogen activator inhibitor 1 (Pai1) and cyclin-dependent kinase inhibitor 1A (CDKN1A) are epigenetically regulated by TGF-β or high-glucose conditions in mesangial cells. Both *Pai1* and *CDKN1A* are pro- fibrotic genes that contribute to DKD pathogenesis by promoting fibrosis, hypertrophy, tubular dysfunction, and glomerulosclerosis. Increased expression of these genes has been associated with enhanced deposition of activating histone marks (H3K9ac and H3K14ac) and the recruitment of the histone acetyltransferases p300 and CREB-binding protein (CBP) at Smad and Sp1 promoters [98].

Hyperglycemia also induces sustained DNMT1 overexpression and histone methylation changes, even after glucose normalization, establishing an epigenetic “memory” [23,99]. Short-term hyperglycemic exposure can trigger long-lasting H3K4 monomethylation at the NF-κB p65 promoter, maintaining pro-inflammatory gene expression and vascular dysfunction through persistent VCAM-1 and CCL-2 expression [23]. These effects are prevented by suppressing mitochondrial superoxide production. Similarly, decreased H3K9me2/3 at the NF-κB promoter sustains its expression, promoting inflammation [50]. Furthermore, hyperglycemia-induced PTMs at nitric oxide synthase (NOX4) and endothelial nitric oxide synthase (eNOS) promoters perpetuate ROS production and vascular damage [100]. Epigenetic enzymes such as histone methyltransferases (HMTs), histone demethylases (HDMs), and DNMTs interact antagonistically to maintain inflammatory gene expression [23,101,102]. The HMT EZH2, a PRC2 complex component, contributes to H3K27me3 deposition on the HIC1 gene in glucose-stimulated tubular endothelial cells, repressing SIRT1, a key antioxidant factor, and promoting ROS accumulation [102].

### 4.2. DNA Methylation Status Connected to Renal Disease Progression

Likewise histone methylation, altered DNA methylation patterns influence the expression of cytokines and chemokines. Hyperglycemia induces dramatic changes in DNA methylation status that coincide with the activation of pathways associated with renal disease progression [13]. Mechanistically, high glucose exposure leads to the accumulation of ROS, which not only causes direct damage to glomerular cells but also alters gene expression via epigenetic changes [13]. Emerging evidence indicates that oxidative stress, a known driver of tubular damage, directly modulates DNA methylation patterns by altering the function of key methylation enzymes, specifically DNMTs and TET proteins [103]. This interplay provides a direct causal mechanism linking environmental stress (oxidative damage) to sustained epigenetic outcomes that drive disease. Changes in DNA methylation were specifically observed in the promoter regions of genes related to oxidative stress, inflammation, and renal fibrosis in both glomerular and tubular cells under diabetic conditions [13].

Studies focusing on advanced CKD patients, particularly those undergoing dialysis, have established a strong relationship between global DNA methylation status in circulating immune cells and poor clinical outcomes [104,105]. Global DNA hypermethylation correlates with systemic inflammation and poor clinical outcomes [106,107]. In contrast to systemic methylation markers, highly localized, tissue-specific methylation changes offer critical information for predicting organ-specific progression. Gluck et al. identified 471 methylation probes associated with renal failure, enriched within kidney regulatory regions, including the epidermal growth factor (EGF) gene locus, capable of predicting renal function decline in DKD [108,109]. Additional evidence indicates that diabetes-related DNA methylation changes in genes regulating oxidative stress, inflammation, and fibrosis occur in glomerular and tubular cells [92]. Similarly, hyperglycemia-induced DNA methylation alterations parallel the activation of pathways driving renal disease progression. Furthermore, extensive alterations in histone methylation have been observed in diabetic kidney tissue, with numerous sites exhibiting both hypermethylation and hypomethylation compared to non-diabetic controls [110]. Complementary evidence from a study published in *Diabetic Medicine* identified distinct DNA methylation patterns in mitochondrial protein-coding genes associated with kidney disease in type 1 diabetes. This study highlighted 51 genes, including TAMM41 and COX6A1, exhibiting both genetic and epigenetic associations with DKD [111]. Moreover, differential methylation of genes essential for mitochondrial metabolism—such as *PMPCB*, *TSFM*, and *AUH*—further implicates epigenetic dysregulation in the pathogenesis of DKD and progression to end-stage renal failure [111].

### 4.3. The Role of Sirtuins in the Hyperglycemia-Induced Epigenetic Memory

Among epigenetic regulators, the NAD^+^-dependent histone deacetylases known as Sirtuins have drawn substantial attention. In particular, SIRT1 regulates mitochondrial energy metabolism and redox balance [112]. Experimental studies demonstrate that SIRT1 activation stimulates the LKB1/AMPK/ROS pathway, suppressing ROS production and inflammation [113]. Conversely, hyperglycemia-induced the Poly (ADP-ribose) polymerase (PARP) activation inhibits SIRT1, forming a self-perpetuating cycle of oxidative stress. Wang and colleagues elucidated a molecular mechanism underlying the protective role of SIRT1. Their study revealed that miRNA-155 expression is markedly elevated in the serum and kidney tissues of diabetic mice, as well as in podocytes exposed to high-glucose conditions. This upregulation enhances the production of inflammatory mediators in podocytes and appears to occur via suppression of SIRT1 mRNA expression [114].

Additionally, SIRT1 also deacetylates endothelial eNOS, maintaining endothelial integrity and preventing vascular senescence and macrophage foam cell formation [115]. Downregulation of SIRT1 elevates miR-34a levels, which enhances p66shc-mediated ROS production and endothelial dysfunction [116]. Overexpression of SIRT1 activates the NRF2/ARE antioxidant pathway, increasing expression of HO-1 and SOD, and reducing AGE-induced ROS in mesangial cells [117].

Other Sirtuins, including SIRT3 and SIRT7, are critical for mitochondrial dynamics and renal protection. SIRT3 mitigates oxidative stress and metabolic dysfunction in proximal tubular cells [118,119]. Reduced SIRT3 expression correlates with increased mitochondrial mtROS and cardiovascular risk [120,121]. The mtROS accumulation activates PARP, leading to endothelial dysfunction—a key event in diabetic complications [122]. SIRT7, responsible for H3K18 deacetylation, plays a protective role against hyperglycemia-induced endothelial-to-mesenchymal transition (EndMT) and podocytes apoptosis [122,123,124,125]. Its downregulation after transient hyperglycemia perpetuates inflammation via the ELK1/DAPK3 axis and enhances vascular injury [126,127]. Notably, miR-20b directly targets SIRT7, and its overexpression induces podocytes apoptosis through caspase-3 activation, whereas SIRT7 restoration mitigates this damage [124].

### 4.4. MicroRNAs and the Epigenetic Memory

MiRNAs have emerged as pivotal epigenetic regulators in hyperglycemia-induced kidney disease, fine-tuning gene expression programs that control fibrosis, inflammation, and apoptosis [128,129]. Yao et al. demonstrated that transient hyperglycemia could induce long-lasting inflammation through the NF-κB/miR-27a-3p/NRF2/ROS/TGF-β/EndMT feedback loop, and the activation of NRF2 can reverse these effects. NRF2, a master regulator of antioxidant defense, controls the expression of genes such as SOD2 and GPX, thereby maintaining cellular redox balance [130]. Its deregulation throughout the stages of CKD reflects an initial compensatory activation followed by functional exhaustion, likely mediated by epigenetic memory mechanisms [131,132]. Aberrant miRNA expression contributes, not only to DKD, but also to other vascular complications, such as atherosclerosis, retinopathy, and microvascular dysfunction [133,134]. Endothelial cells miRNA profiling revealed that miR-130b-3p, miR-221-3p, and miR-140-5p are glucose-responsive miRNAs that target genes involved in apoptosis and angiogenesis [135]. Persistent upregulation of miR-125b activates NF-κB signaling, while reduced miR-146a-5p enhances pro-inflammatory gene transcription [136]. Similarly, miR-214 downregulates PTEN, promoting Akt/mTORC activation and renal injury [137], whereas miR-22 suppresses PTEN and autophagy, thereby driving fibrotic responses [138]. Under hyperglycemic conditions, miR-21 also suppresses PTEN while activating Akt and TORC1 pathways; its overexpression, commonly observed in DKD, contributes to renal cell hypertrophy, fibrosis, and EMT [139]. Dysregulation of this and other miRNAs, including members of the miR-200 family, constitutes a central mechanism in DKD progression. The increased TGF-β expression observed in diabetic nephropathy is mediated by miR-192, which in turn regulates miR-200, further amplifying glomerular fibrogenesis and hypertrophy [140]. The crucial role of miR-126 in endothelial protection and angiogenesis has positioned it as a major focus of research in both the pathophysiology of diabetes and its vascular complications [141]. Notably, patients with CKD exhibit reduced levels of the atheroprotective, endothelium-derived miR-126, and its downregulation has been specifically associated with both micro- and macrovascular complications in type 1 diabetes [142]. In addition, circulating miRNA profiles are significantly altered in response to hyperglycemia. In patients with type 1 diabetes, miR-125b-5p and miR-365a-3p levels shown a positive correlation with HbA1c levels, whereas miR-5190 and miR-770-5p display negative correlations [143]. Furthermore, extracellular vesicles enriched with miR-15 and miR-16 derived from diabetic patients have been shown to preserve a “metabolic memory” signal that perpetuates endothelial CaMK2a activation and cardiac dysfunction [144].

The ongoing elucidation of epigenetically regulated networks in CKD is providing deeper insights into disease pathogenesis and potential therapeutic strategies. Sustained endothelial dysfunction—driven by oxidative stress, inflammation, EndMT, and disrupted metabolic homeostasis—is maintained by epigenetic “memory” mechanisms that stabilize pathological phenotypes [145] (Figure 3). Consequently, effective therapeutic approaches for CKD should extend beyond glycemic control to specifically target molecular and epigenetic memory pathways, offering new opportunities to halt or even reverse renal functional decline.

## 5. Epigenetic Players as Therapeutic Targets and Biomarkers for CKD Patients’ Stratification

Epigenetic molecules act cooperatively to preserve the integrity of the epigenetic machinery and maintain cell identity. They can be categorized functionally as: (i) Writers, which add chemical marks (e.g., methyl or acetyl groups) to DNA or histones, including DNMTs, HATs, and HMTs; (ii) Readers, which recognize and bind specific epigenetic marks, such as bromodomains (acetyl-lysine readers) and chromodomains (methyl-lysine readers); (iii) Erasers, which remove post-translational modifications, including HDACs and HDMs [146,147].

The therapeutic targeting of these regulators has emerged as a promising pharmacological avenue. Several epigenetic modulators—collectively known as epidrugs—have already achieved clinical success in oncology. DNMT inhibitors (DNMTi; e.g., azacytidine, decitabine) and HDAC inhibitors (HDACi; e.g., vorinostat, panobinostat) have been approved for the treatment of myelodysplastic syndromes, leukemias, and other hematologic malignancies [147,148,149,150]. The current landscape of FDA-approved epidrugs includes a wide range of compounds (Table 2), and numerous inhibitors (e.g., ivaltinostat, AR-42, abexinostat, bisthianostat, valproic acid) are undergoing clinical evaluations.

Combination strategies integrating epidrugs with traditional chemotherapeutics are being explored to curb both solid and hematologic tumor progression [151,152,153,154]. Beyond oncology, epigenetic modulators are being investigated for a variety of conditions, including cardiovascular, neurodevelopmental, and neurodegenerative disorders [155,156]. Originally designed for cancer, these compounds also exhibit potential for mitigating chronic inflammation, oxidative stress, and fibrotic remodeling—key mechanisms underpinning metabolic memory and CKD progression [157,158,159,160]. Because epigenetic marks are reversible, targeting the enzymes that write, read, and erase them, including through miRNA mimics or inhibitors, offers a promising therapeutic strategy for managing CKD and its complications, and supports the development of precision medicine approaches for high-risk patients.

### 5.1. Epigenetic Biomarkers in CKD and Diabetes

Early biomarker identification is critical for timely diagnosis, intervention, and prevention of progression to ESRD. Epigenome-wide association studies (EWAS) in individuals newly diagnosed with diabetes have uncovered blood-based epigenetic biomarkers capable of: (i) early risk stratification, (ii) predicting CKD development, and (iii) monitoring disease progression. To elucidate the relationship between metabolic memory and epigenetic modifications, several genome-wide studies based on samples from the DCCT/EDIC cohort have been conducted [161]:(1)In a first study, patients exhibited elevated levels of the active chromatin mark H3K9ac in monocytes, correlated with baseline hyperglycemia and the upregulation of genes linked to inflammation and vascular complications. H3K9ac thus emerged as a potential biomarker for metabolic memory [162].(2)In a second study, DNA methylation profiling at two time points revealed 12 persistently differentially methylated loci, notably TXNIP, which was hypomethylated in patients with more complications [163]. Other validations confirmed a strong inverse correlation between TXNIP methylation and glycemic control, suggesting that TXNIP hypomethylation reflects chronic hyperglycemia and may predict renal and vascular damage [164].(3)In a third study, longitudinal analysis (18-year follow-up) showed that specific HbA1c-associated CpGs predicted DKD risk, again highlighting TXNIP as a key biomarker [165,166].

These findings, corroborated by studies across tissues such as kidney, nerves, and retina, suggest TXNIP as a pivotal player in metabolic memory and a target for intervention [97,167,168].

A longitudinal study from Lund University followed 752 newly diagnosed type 2 diabetes patients, identifying over 400 differentially methylated sites associated with cardiovascular events, 87 of which were used to build a methylation risk score for cardiovascular risk prediction [169]. Similar EWAS have identified differential methylation in genes regulating mitochondrial function, fibrosis, and inflammation, molecular hallmarks of persistent renal injury [14,111,170,171,172].

A methylation score proposed by Marchiori et al., derived from 37 methylation sites, predicted CKD incidence during an 11.5-years follow-up [173]. Moreover, aberrant DNA methylation patterns in key genes (mTOR, RPTOR, IRS2, GRK5, LCAT, SLC27A3, SLC1A5) have been linked to DKD progression [174]. Elevated DNMT1 expression in DKD mononuclear cells induces mTOR-related methylation changes and inflammation, reversible through DNMT1 inhibition with 5-aza-2′-deoxycytidine (5-aza), offering a new therapeutic perspective [175].

Different compounds have been proven to act on the key mechanisms of hyperglycemic memory, such as epigenetic modifications, inflammation, and senescence, but none have been approved for CKD and complications. The DNMT1 inhibitor SGI-1027 disrupts p21 methylation, attenuating senescence and fibrosis in DKD models [45,176]. Histone methylation also contributes to metabolic memory. Downregulation of the demethylase JMJD3 (also called KDM6B) correlates with neointimal hyperplasia in CKD-related vascular lesions [177]. JMJD3 loss promotes EndMT, vascular remodeling, and nitric oxide deficiency. Additionally, EZH2 upregulation, with concurrent loss of JMJD3, represses antioxidant genes (SOD1/2, JunD), exacerbating oxidative stress [178]. In fact, its upregulation is linked to tubular cell injury, podocytes dedifferentiation, fibroblast proliferation, inflammatory cytokines/chemokines production and infiltration of inflammatory cells [179]; moreover, depleting EZH2 from podocytes culture in high-glucose conditions derepressed the endogenous antioxidant TXNIP, abrogating ROS accumulation [180]. These findings make EZH2 as a legitimate therapeutic target; however, no current clinical trials of EZH2 inhibitors are available for treating hyperglycemia-induced CKD.

Aberrant activity of BET (bromodomain and extra-terminal domain) proteins—particularly BRD2, BRD3, and BRD4—also sustains vascular and renal injury. BET inhibitors (BETi) such as JQ1 and Apabetalone (RVX-208) have shown preclinical efficacy in DKD and diabetic cardiomyopathy. The selective pharmacological inhibition of BRD4 with JQ1 restored mitochondrial functions via the PTEN-induced PINK1/Parkin mitophagy pathway [181,182]. Besides, the phase 3 BETonMACE trial (ClinicalTrials.gov ID: NCT02586155) tested Apabetalone in type 2 diabetes, with or without CKD, focusing on cardiovascular complications. While it did not significantly reduce major adverse cardiac events overall, it markedly reduced heart failure hospitalizations, particularly among CKD patients’ subgroup [183]. Further clinical studies are needed to confirm these brilliant results in a larger patients’ sample; nevertheless, future clinical settings must be adjusted to reduce side effects [184].

### 5.2. Histone Deacetylase Inhibitors and miRNA-Based Therapies

HDAC inhibitors are gaining interest in diabetic nephropathy [50,185]. Non-selective inhibitors such as Vorinostat (SAHA) improved renal function and reduced fibrosis in animal models [186]. Another HDACi Valproic acid (VPA), used for epilepsy and neurologic disorders, was demonstrated to repress the NF-κB pathway, inactivating inflammation and alleviating podocytes and renal cell injury [187]. The same effects on inflammation and fibrosis have been observed with the HDACi trichostatin A (TSA); this compound was able to influence the metabolic memory through the inhibition of TGF-β1, the decrease in extracellular matrix deposition and the escape from the phenomenon of EMT [188]. SIRT1, a class III HDAC, regulates metabolic memory by deacetylating p53 and modulating apoptosis, inflammation, and oxidative stress [113,189]; its activity is negatively regulated by miR-200a-3p, whose inhibition mitigates tubular injury, suggesting its biomarker potential for diabetes- and hypertension-related nephropathy [190].

MiRNAs are both biomarkers and therapeutic targets in DKD. Two strategies have emerged:Inhibition of overexpressed miRNAs (via antisense oligonucleotides, gene knockouts, or “sponges”);Restoration of protective miRNAs (via double-stranded mimics or expression vectors).

Examples include (i) miR-29b mimics, which interfere with Sp1-dependent, TGF-β/Smad3-mediated renal fibrosis, NF-κB–driven renal inflammation, and Th1-associated immune injury [191,192]; and (ii) inhibitors of miR-21 and miR-192, which reduce albuminuria and renal fibrosis, maintaining the structural and functional integrity of the kidney [193,194]. Circulating miRNAs such as miR-1, miR-133a, and miR-126 serve as non-invasive biomarkers for diabetic cardiomyopathy and cardiovascular risk [195,196].

In a study by Al-Kafaji et al., diabetic nephropathy was associated with upregulation of miR-377 and downregulation of miR-192, correlating with albuminuria severity and identifying miR-377 as a positive, and miR-192 as a negative, biomarker of DKD [197]. Recent approaches advocate miRNA panels (e.g., miR-21, miR-34a, miR-133a) to enhance diagnostic and prognostic power, bridging therapy and diagnostics—theranostics—for patient stratification [14,198].

### 5.3. Integrating Machine Learning and Multi-Omics

Recent advances in Artificial Intelligence (AI) have enabled integration of epigenetic, genomic, transcriptomic, and metabolomic data to predict disease risk and uncover novel therapeutic targets. A recent study from the Middle East that combine EWAS, genome-wide analyses, and metabolomics identifies several previously unrecognized methylated genes, genomic variants, and metabolic pathways implicated in diabetes [199]. These integrative analyses revealed clinically relevant molecular subnetworks, offering new avenues for the development of translational multi-omics applications.

Similarly, another multi-omics investigation integrating metabolomics, transcriptomics, proteomics, and epigenomics identified the transcription factor KLF15 as a critical regulator of diabetic cardiomyopathy, positioning it as a powerful therapeutic target [200].

The application of machine learning algorithms represents a paradigm shift in the management of diabetes and its cardiovascular complications. A compelling example is the Multi-Ethnic Study of Atherosclerosis (MESA) clinical trial (ClinicalTrials.gov ID: NCT00005487), in which machine learning-based modeling of protein–protein interaction networks, when combined with imaging tomography and deep phenotyping, successfully predicted cardiovascular outcomes [201]. Coupling machine learning algorithms with epigenomics and multi-omics datasets can further facilitate the discovery of novel biomarkers for cardiovascular risk stratification in diabetic patients. This integrated approach holds the potential to transform patient care by enabling earlier and more precise diagnoses, individualized risk assessment, and targeted treatment interventions.

## 6. Conclusions

Further studies are urgently needed to validate candidate biomarkers and identify safer, more effective epigenetic drugs. Integrating multi-omics data will allow stratification of patients predisposed to vascular complications and enhance precision medicine in CKD. Understanding the dynamic role of epigenetic regulation is pivotal for predicting disease trajectory, preventing toxin-induced CKD progression, and overcoming micro- and macrovascular complications.

## Figures and Tables

**Figure 1 genes-16-01442-f001:**
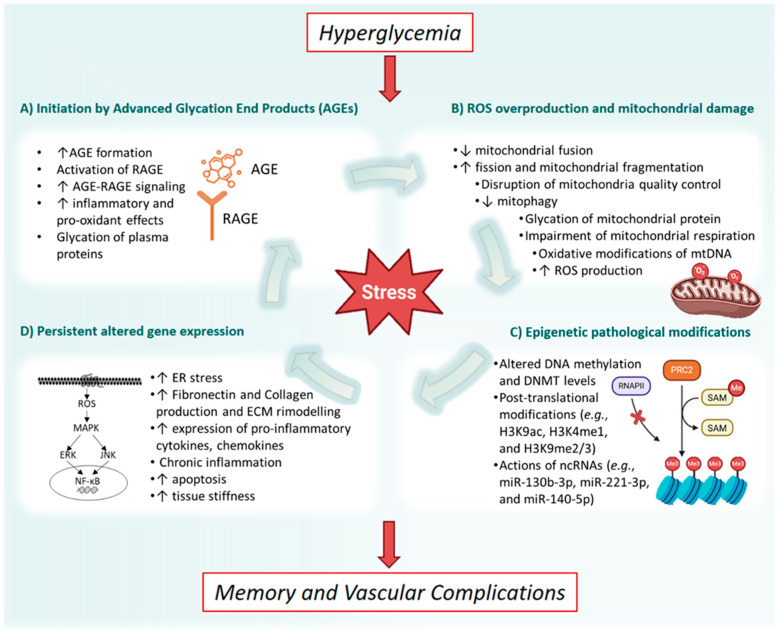
Molecular pathophysiology of hyperglycemic memory in the development of vascular complications. The process is principally driven by four arms. (**A**) Initiation by formation of Advanced Glycation End Products: hyperglycemia promotes the non-enzymatic formation of advanced end products, followed by the binding to their specific receptor, subsequently activating a cascade of pro-inflammatory and pro-oxidant signals; (**B**) ROS overproduction and mitochondrial damage: mitochondrial dysfunctions, characterized by increased fission, decreased fusion, impaired quality control via mitophagy, and compromised respiratory efficiency are induced and result in the overproduction of reactive oxygen species; (**C**) Epigenetic pathological modification: the cumulative cellular stress from AGE overproduction and excessive reactive oxygen species induce an epigenetic reprogramming, including alteration of DNA methylation, aberrant post-translational histone modifications, and dysregulation of non-coding RNAs; (**D**) Persistent altered gene expression: transient or sustained hyperglycemia induces durable cellular and epigenetic reprogramming characterized by endoplasmic reticulum stress and the chronic activation of transcription factors, which drives the expression of pro-inflammatory mediators, promotes extracellular matrix remodeling, and increases endothelium stiffness. AGE: Advanced Glycation End Products; RAGE: Receptor for Advanced Glycation End Products; mtDNA: mitochondrial DNA; ROS: reactive oxygen species; DNMT: DNA methyltransferases; ncRNA: non-coding RNA; ER: endoplasmic reticulum; ECM: extracellular matrix; ↓: increase; ↑: reduction. Created with Biorender.com.

**Figure 2 genes-16-01442-f002:**
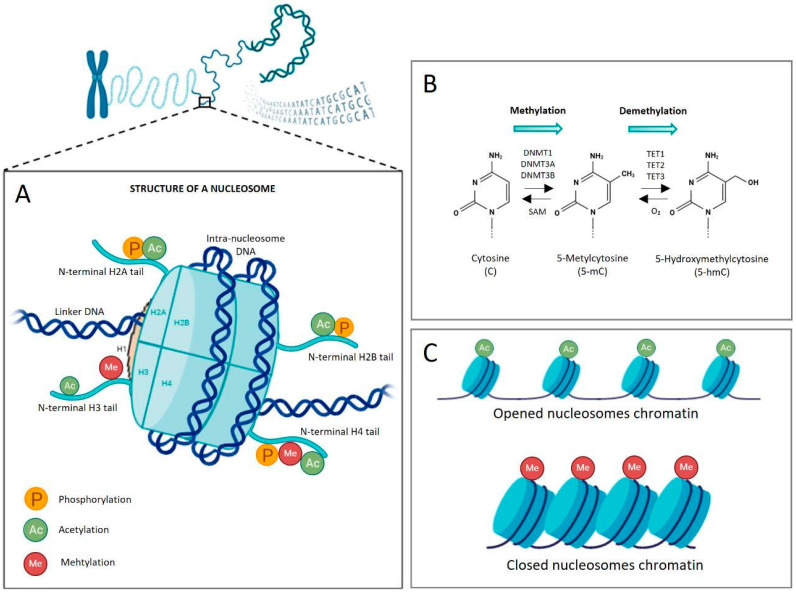
Schematic representation of the organization and packaging of elements of the nucleosome. (**A**) Nucleosome as the fundamental unit of the epigenome: the structure of the nucleosome consists of an octamer of core histone proteins (two copies each of H2A, H2B, H3, and H4) and the H1 linker histone, around which DNA is wrapped. Histone tails undergo post-translational modifications at different amino acid residues and biochemical groups added at the N-terminal region are represented (e.g., acetyl, methyl, and phosphoryl group); (**B**) Dynamics of DNA methylation: enzymatic processes governing DNA methylation and demethylation. DNA methyltransferases (DNMT1, DNMT3A, DNMT3B) catalyze the addition of a methyl group to the 5′ position of cytosine, forming 5-methylcytosine (5-mC); conversely, the ten-eleven translocation family of dioxygenases initiates active demethylation by oxidizing 5-mC to 5-hydroxymethylcytosine; (**C**) Chromatin architecture and function as a consequence of epigenetic marks deposition: covalent modifications of histone tails can shape chromatin conformation, impacting on gene expression. The acetylation of lysine residues break down the affinity between positive charge of lysines and negative charge of DNA, inducing relaxation of the chromatin structure and facilitating gene transcription; conversely, DNA methylation enhances affinity between DNA and histones, resulting in condensed chromatin (in form of heterochromatin) and inactivation of the transcription. SAM: S-adenosyl-methionine; TET: ten-eleven translocation. Created with Biorender.com.

**Figure 3 genes-16-01442-f003:**
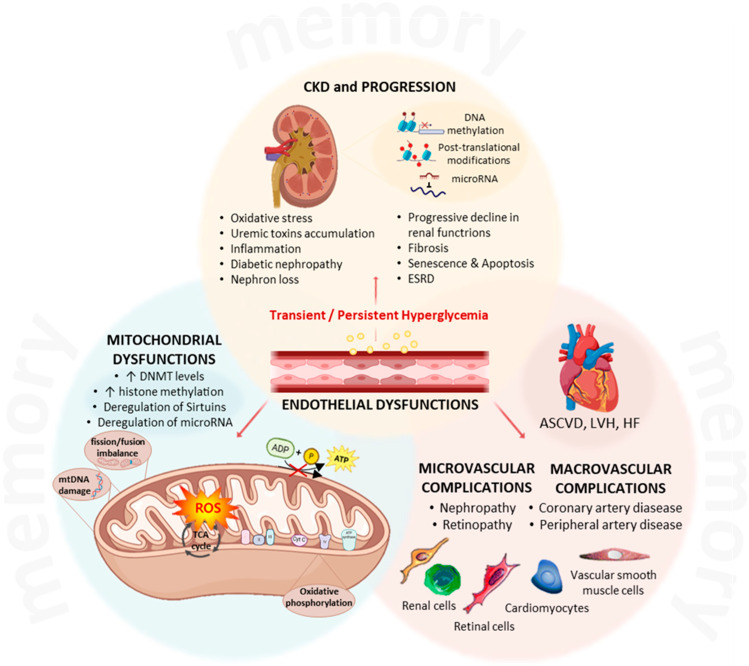
The central role of hyperglycemia-induced changes in driving the progression of chronic kidney disease. Transient or persistent hyperglycemia acts as the *primum movens* to establish the memory phenomenon and to initiate the cascade of downstream pathological events, centered around mitochondrial, endothelial, and vascular dysfunctions. At the cellular level, high glucose impacts on mitochondrial functions, inducing overproduction of reactive oxygen species through oxidative phosphorylation, resulting in fission/fusion imbalance and damage to the mitochondrial DNA. These processes are regulated by an epigenetic memory, able to create changes in DNA methylation, post-translational modification (e.g., Sirtuin activation/repression), and miRNA deregulations. Hyperglycemia also promotes endothelial dysfunctions, with activation of inflammatory pathways with involvement of molecular and epigenetic targets, resulting in imbalance in vascular homeostasis. The initial stimulus, coupled with endothelial and mitochondrial dysfunction, triggers a vicious cycle within the kidney, producing oxidative stress, accumulation of nephrotoxins, and chronic inflammation. These factors could contribute to the progression of chronic kidney disease, with fibrosis, senescence, apoptosis, ultimately predisposing to the onset of end-stage renal disease. Moreover, the systemic nature of endothelial dysfunction gives rise to widespread vascular complications, classified in microvascular and macrovascular complications, such as retinopathy and artery disease. The dysfunction of cardiomyocytes and of vascular smooth muscle cells can be involved, finally, in structural damages and adverse cardiac outcomes, such as atherosclerotic cardiovascular disease, left ventricular hypertrophy, and heart failure. CKD, chronic kidney disease; ESRD, end-stage renal disease; DNMT, DNA methyltransferase; mtDNA, mitochondrial DNA; ROS, reactive oxygen species; TCA, tricarboxylic acid; ASCVD, atherosclerotic cardiovascular disease; LVH, left ventricular hypertrophy; HF, heart failure; ↓: increase; ↑: reduction. Created with Biorender.com.

**Table 1 genes-16-01442-t001:** HDAC enzymes classes.

Class	Enzymes	Common Name/Characteristics	Mechanism
I	HDAC1, HDAC2, HDAC3, HDAC8	“Classical” HDACs	Traditional deacetylation mechanism
II	HDAC4, HDAC5, HDAC6, HDAC7, HDAC9, HDAC10	“Classical” HDACs	Traditional deacetylation mechanism
III	SIRT 1–7	Sirtuins	NAD^+^-dependent mechanism
IV	HDAC11	“Classical” HDACs	Shares only weak homology with Class I and II

HDAC, histone deacetylases; SIRT, silent information regulator Sirtuin; NAD^+^, nicotinamide adenine dinucleotide.

**Table 2 genes-16-01442-t002:** List of FDA-approved epigenetic drugs.

Drug Name	Target	Indication	Year
Azacitidine (Vidaza^®^)	DNMTi	Myelodysplastic syndromes, Chronic myelomonocytic leukemia, Acute myeloid leukemia	2004 2004 2007
Decitabine (Dacogen^®^) Decitabine-Cedazuridine (Inqovi^®^)	DNMTi	Myelodysplastic syndromes, Intermediate/high-risk myelodysplastic syndromes	2006 2020
Valproic Acid (Depakin^®^)	Class I/II HDAC	Epilepsy, bipolar disorder, migraine	2010
Vorinostat (Zolinza^®^)	Class I/II HDAC	Cutaneous T-cell lymphoma	2006
Romidepsin (Istodax^®^)	HDAC6	Cutaneous T-cell lymphoma, Peripheral T-cell lymphoma (withdrawn 2021)	2009 2011
Belinostat (Beleodaq^®^)	Non-selective HDACi	Relapsed or refractory peripheral T-cell lymphoma	2014
Panobinostat (Farydak^®^)	Non-selective HDACi	Relapsed multiple myeloma (discontinued 2021)	2015
Tazemetostat (Tazverik^®^)	EZH2i	Metastatic or locally advanced epithelioid sarcoma, Relapsed or refractory follicular lymphoma	2020 2020

DNMT, DNA methyltransferase; HDAC, histone deacetylases; EZH2, enhancer of zeste homolog 2.

## Data Availability

The original contributions presented in this study are included in the article. Further inquiries can be directed to the corresponding author.

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
