# Peer review of "From Metabolic to Epigenetic Memory: The Impact of Hyperglycemia-Induced Epigenetic Signature on Kidney Disease Progression and Complications"

_genes, 2025, doi:10.3390/genes16121442_

Round 1
Reviewer 1 Report
Comments and Suggestions for Authors
The manuscript reviews the impact of hyperglycemia-induced epigenetic signatures—particularly metabolic and epigenetic memory—on chronic kidney disease progression and complications, with emphasis on mechanisms such as DNA methylation, histone modifications, and non-coding RNAs.
While the review by Cannito et al. is well-written and highly relevant, the integration of DKD pathology is completely lacking.
The three main pathological lesions of diabetic nephropathy are diffuse mesangial cell expansion, GBM thickening, and arteriolar hyalinisation. However, almost all kidney compartments, including the glomerular capillary wall, podocytes, mesangium, tubulointerstitium and renal vasculature, are affected (Rout & Jialal, Diabetic Nephropathy. Diabetic Nephropathy). StatPearls, 2025, https://www.ncbi.nlm.nih.gov/books/NBK534200/). See also Tervaert et al., 'Pathologic Classification of Diabetic Nephropathy'. JASN 21(4): p. 556–563, April 2010, https://doi.org/10.1681/ASN.2010010010.
I suggest to add a discussion of DKD pathology and known epigenetic changes.
See e.g.
- Kuo et al. Int. J. Mol. Sci. 2022, 23(2), 843; https://doi.org/10.3390/ijms23020843
- Zhuo et al. Update: the role of epigenetics in the metabolic memory of diabetic complications. American Journal of Physiology-Renal Physiology 2024 327:3, F327-F339, https://doi.org/10.1152/ajprenal.00115.2024
Author Response
Reviewer’s Comment: The manuscript reviews the impact of hyperglycemia-induced epigenetic signatures - particularly metabolic and epigenetic memory - on chronic kidney disease progression and complications, with emphasis on mechanisms such as DNA methylation, histone modifications, and non-coding RNAs.
While the review by Cannito et al. is well-written and highly relevant, the integration of DKD pathology is completely lacking.
The three main pathological lesions of diabetic nephropathy are diffuse mesangial cell expansion, GBM thickening, and arteriolar hyalinisation. However, almost all kidney compartments, including the glomerular capillary wall, podocytes, mesangium, tubulointerstitium and renal vasculature, are affected (Rout & Jialal, Diabetic Nephropathy. Diabetic Nephropathy. StatPearls, 2025, https://www.ncbi.nlm.nih.gov/books/NBK534200/). See also Tervaert et al., 'Pathologic Classification of Diabetic Nephropathy'. JASN 21(4): p. 556–563, April 2010, https://doi.org/10.1681/ASN.2010010010. I suggest to add a discussion of DKD pathology and known epigenetic changes. See e.g. Kuo et al. Int. J. Mol. Sci. 2022, 23(2), 843; https://doi.org/10.3390/ijms23020843; Zhuo et al. “Update: the role of epigenetics in the metabolic memory of diabetic complications”. American Journal of Physiology-Renal Physiology 2024 327:3, F327-F339, https://doi.org/10.1152/ajprenal.00115.2024.
Authors’ Response: We are grateful for the valuable suggestion to expand in further detail on DKD pathology into our review. We agree that discussing the key pathological lesions is essential for connecting the epigenetic mechanisms we describe with the consequence of the disease.
In response to the insightful recommendation, we have carefully revised the manuscript. We have now incorporated a detailed discussion of the main pathological hallmarks of DKD throughout the text, linking these structural changes to the specific epigenetic signatures we review, including DNA methylation, histone modifications, and the role of non-coding RNAs in this context.
We are confident that, by weaving these pathological concepts throughout the manuscript, we have not only addressed the reviewer’ concern but have also significantly strengthened our work. The text now provides a more cohesive narrative that bridges the gap between molecular epigenetic alterations and the structural damage observed in diabetic kidney disease. Specifically, we have now made the following additions through the manuscript:
Line 70: Diabetic kidney disease (DKD) is a major microvascular complication of both type 1 and type 2 diabetes and a leading cause of ESRD worldwide [7]. DKD encompass a series of structural and functional changes, including podocyte effacement, mesangial expansion, excessive deposition of extracellular matrix (ECM), and tubular epithelial-to-mesenchymal transition (EMT) [8]. Alongside hypertension, high glucose level is the major pathological cause of DKD [9].
Line 83: Epigenetic dysregulation affects various renal cell types, including mesangial cells, podocytes, tubular epithelia, and glomerular endothelial cells [14]. These alterations, which do not change the DNA sequence itself, modify gene expression patterns, leading to cellular damage and fibrosis, typical hallmarks of DKD. These environmentally-responsive mechanisms may mediate the sustained expression of genes and phenotypes associated with DKD.
Line 147: Collectively, these ROS-associated pathways contribute to glomerular cell dysfunction, renal inflammation, and fibrosis by promoting DNA damage, mitochondrial impairment, lipid peroxidation, and abnormal protein modifications [17].
Line 154: Interestingly, transient or prolonged hyperglycemic insults can leave a persistent molecular fingerprint that continues to drive diabetic complications, like DKD, even after the normoglycemia is restored [22-26].
Line 411: DKD represents a paradigmatic example in which sustained hyperglycemia induces profound epigenetic alterations - such as DNA methylation and histone modifications - that reprogram pathological gene expression [91,92]. Notably, histone modifications observed in the proximal tubular cells of diabetic mice persist even after normalization of glycemia, underscoring the long-lasting nature of these epigenetic changes [93,94].
Line 434: Preclinical evidence from DKD models also indicates that genes such as plasminogen activator inhibitor 1 (Pai1) and cyclin-dependent kinase inhibitor 1A (CDKN1A) are epigenetically regulated by TGF-β or high-glucose conditions in mesangial cells. Both Pai1 and CDKN1A are pro- fibrotic genes that contribute to DKD pathogenesis by promoting fibrosis, hypertrophy, tubular dysfunction, and glomerulosclerosis. Increased expression of these genes has been associated with enhanced deposition of activating histone marks (H3K9ac and H3K14ac) and the recruitment of the histone acetyltransferases p300 and CREB-binding protein (CBP) at Smad and Sp1 promoters [100].
Line 484: Furthermore, extensive alterations in histone methylation have been observed in diabetic kidney tissue, with numerous sites exhibiting both hypermethylation and hypomethylation compared to non-diabetic controls [113]. Complementary evidence from a study published in Diabetic Medicine identified distinct DNA methylation patterns in mitochondrial protein-coding genes associated with kidney disease in type 1 diabetes. This study highlighted 51 genes, including TAMM41 and COX6A1, exhibiting both genetic and epigenetic associations with DKD [114]. Moreover, differential methylation of genes essential for mitochondrial metabolism - such as PMPCB, TSFM, and AUH - further implicates epigenetic dysregulation in the pathogenesis of DKD and progression to end-stage renal failure [114].
Line 501: Wang and colleagues elucidated a molecular mechanism underlying the protective role of SIRT1. Their study revealed that miRNA-155 expression is markedly elevated in the serum and kidney tissues of diabetic mice, as well as in podocytes exposed to high-glucose conditions. This upregulation enhances the production of inflammatory mediators in podocytes and appears to occur via suppression of SIRT1 mRNA expression [117].
Line 533: Aberrant miRNA expression contributes, not only to DKD, but also to other vascular complications, such as atherosclerosis, retinopathy, and microvascular dysfunction [136,137].
Line 541: Under hyperglycemic conditions, miR-21 also suppresses PTEN while activating Akt and TORC1 pathways; its overexpression, commonly observed in DKD, contributes to renal cell hypertrophy, fibrosis, and EMT [142]. Dysregulation of this and other miRNAs, including members of the miR-200 family, constitutes a central mechanism in DKD progression.
Final Remarks
We sincerely thank reviewers for their thoughtful and detailed feedback. The revisions have notably improved the manuscript’s clarity, structure, and scientific rigor. We believe the current version addresses all concerns raised and is now suitable for publication in Genes.
Reviewer 2 Report
Comments and Suggestions for Authors
The authors present an interesting and detailed review examining the impact of chronic hyperglycaemic conditioning on kidney disease initiation and progression. Briefly, the authors focus on the aspects pertaining to epigenetic regulation of the DNA code, exploring how the various mechanisms which interact with and influence genetic and downstream signalling are altered in the context of a hyperglycaemic environment. The review is detailed, giving a good scope and depth in the content while also examining therapeutic advancements designed towards such. Overall, this was an informative read, though I did note some aspects that could be addressed. The authors should consider the following when preparing a suitable revision.
- The writing is clear for the most part, but the language could be refined/simplified in the interest of the flow of the piece. There are several instances throughout where the writing is overcomplicated through the language used, and the authors should revise the entire piece with this mind and address such.
- Similarly, the structure of the review should be revised. At times, there are a lot of topics discussed under a single heading. Sometimes these are separated from one another by section breaks, whereas for others the content is delivered as one dense block of text. The authors should consider using subheadings/sections in these instances which will improve the flow/accessibility of the content to readers. This is done in Section 4 for example and works very well.
- The headings should be revised with some starting with lower case lettering.
- Figure 1 is good as a concept, but I believe the formatting could be improved. The resolution is poor with some diagrams too small/pixelated to interpret properly, while the writing is too small for the most part. Other Figures such as Figure 4 also suffer from the same. The authors should revise the presentation of the figures in any resubmission.
- The Figures should be moved closer to their mentions in the text. At times there is a great body of text separating them from one another.
- The Figure legends for Figure 2 is separated from the figure itself. Same with Table 2.
Details on the writing can be found in the main body of the report submitted.
Author Response
Reviewer’s General Comment: The authors present an interesting and detailed review examining the impact of chronic hyperglycemic conditioning on kidney disease initiation and progression. [...] The writing is clear for the most part, but the language could be refined/simplified. The structure should also be revised, as several topics appear under one heading. Consider using subheadings, as in Section 4, to improve flow.
Authors’ Response: We thank the reviewer for this realistic and constructive assessment, which we fully share. In response, we simplified complex sentences throughout the text to enhance readability and introduced additional sub-sections and sub-paragraphs where needed. By aligning the structure of the entire manuscript with that of Section 4, the overall organization and navigability have been markedly improved.
Reviewer’s Comment 1: Headings: Some begin with lowercase lettering.
Authors’ Response: We appreciate this observation. The entire manuscript has been reviewed, and all inconsistencies in the formatting of section headings have been corrected.
Reviewer’s Comment 2: Figures: Figure 1 is conceptually good, but the resolution and readability (font size and diagram clarity) should be improved. Similar issues are present in Figure 4.
Authors’ Response: We agree with this valuable point. All figures have been reformatted to improve their resolution, and the text and graphical elements have been enlarged for better clarity. We hope the updated figures now meet the reviewer’s expectations.
Reviewer’s Comment 3: Figure placement: Figures should be moved closer to where they are mentioned in the text.
Authors’ Response: We appreciate this practical suggestion. All figures have been repositioned to appear in close proximity to their first mention, improving structural clarity and readability.
Reviewer’s Comment 4: Figure and table legends: The legend for Figure 2 and Table 2 are separated from their respective items.
Authors’ Response: We are grateful for this detailed observation. All figures and tables are now presented on the same page as their corresponding legends, ensuring proper alignment and consistency.
Final Remarks
We sincerely thank reviewers for their thoughtful and detailed feedback. The revisions have notably improved the manuscript’s clarity, structure, and scientific rigor. We believe the current version addresses all concerns raised and is now suitable for publication in Genes.
Reviewer 3 Report
Comments and Suggestions for Authors
Sustained hyperglycemia initiates a cascade of deleterious molecular, cellular events and epigenetic alterations that collectively contribute to the progression of renal injury. In the current review the authors presented the current knowledge on metabolic and epigenetic mechanisms, with a particular focus on the epigenetic aspects, concerning this pathology. They underlined that, given their reversible nature, epigenetic determinants are emerging as promising biomarkers and therapeutic avenue.
In my opinion, the manuscript topic is interesting and well written. I recommend the manuscript for publication after a minor revision. Some suggestions:
1.Lines 49-51, you wrote “Conversely, diabetes mellitus and hypertension themselves are the primary causes of CKD, accounting for approximately 60% of all cases” For which year is this statistical information valid? and where, worldwide? Please add/clarify.
2.Lines 180-183, you wrote: “Targeting these mitochondrial quality control pathways and metabolic memory regulatory mechanisms, therefore, represents a highly promising avenue for novel therapeutic interventions to mitigate or prevent the relentless progression of the disease and their complications”. Add please more details.
- Please add the origin of the figures.
4.Lines 204-206, you stated: “In fact, intermittent episodes can be significantly more pernicious than sustained and chronic high glucose levels”. Add please more details related with this aspect”. You wrote about this at pages 5-6, but in my opinion more detail will be welcome.
5.Lines 365-366, you wrote: “Currently, a lot of miRNAs are in the spotlight for their role as biomarkers for diagnosis and prognosis….”Add please some examples of involved miRNAs.
- Page 12, Point 4.5. In my opinion also other miRNA are involved in hyperglycemia. Please add about them.
- The information presented at point 5.3 Integrating Machine Learning and Multi-Omics are brief. Please enlarge de section 5.3
Author Response
Reviewer’s Comment 1: Lines 49–51: “Conversely, diabetes mellitus and hypertension themselves are the primary causes of CKD, accounting for approximately 60% of all cases.” Please clarify the year and geographical scope of this statistic.
Authors’ Response: We thank the reviewer for this pertinent observation. The text has been revised to specify the year and the global scope of the statistic, supported by the updated reference. The variation consists in the following:
Line 50: Globally, in 2017, the Global Burden Disease, Injuries, and Risk Factors (GBD) study showed that impaired fasting plasma glucose and high glucose pressure account for 57,6% and 43,2% of the age-standardized rate of CKD, respectively [4].
Reviewer’s Comment 2: Lines 180–183: “Targeting these mitochondrial quality control pathways…” Please add more details.
Authors’ Response: We fully agree. We have expanded this section to include additional discussion on pharmacological approaches targeting mitochondrial dysfunction in endothelial cells under hyperglycemic conditions.
Line 214: Pharmacological agents such as mitochondrial division inhibitor 1 (Mdivi-1) and leflunomide, have shown efficacy in preserving mitochondrial structure and function in endothelial cells [37,47]. Notably, leflunomide administered during the hyperglycemia prevented mitochondrial damage, while post-hyperglycemia treatment restored mitochondrial dynamics, reactivated the mitophagy machinery, reduced ROS production, and inhibited apoptosis [37].
Reviewer’s Comment 3: Please add the origin of the figures.
Authors’ Response: We had already acknowledged BioRender.com in the Acknowledgments section as indicated by the software. Following this suggestion, we have now also included the figure source in each individual caption for transparency.
Reviewer’s Comment 4: Lines 204–206: “In fact, intermittent episodes can be significantly more pernicious than sustained and chronic high glucose levels.” Please elaborate.
Authors’ Response: We appreciate this suggestion. Paragraph 2.3 has been substantially revised to elaborate on this point, including discussion of the deleterious effects of glycemic oscillations on oxidative stress and endothelial dysfunction. The review by Brownlee and Hirsch has been added as reference 49 to support this expansion.
Reviewer’s Comment 5: Lines 365–366: “Currently, a lot of miRNAs are in the spotlight…” Please add examples.
Authors’ Response: In response, we have expanded paragraph 3.6 at line 365 to include specific examples of microRNAs implicated as biomarkers and therapeutic targets in metabolic and renal diseases. Corresponding references were added.
Line 367: For instance, several circulating miRNAs are dysregulated in the blood of cancer patients’, serving as potential biomarkers for early cancer diagnosis [40]. Given their established role in carcinogenesis, panels of miRNAs have been proposed as valuable tools for cancer screening and classification. For example, a specific five-miRNA signature - comprising miR-486-5p, miR-451, miR-92a, miR-25, and miR-16 - has been identified for the early detection of gastric cancer [40].
Reviewer’s Comment 6: Page 12, Point 4.5: Please add more miRNAs involved in hyperglycemia.
Authors’ Response: Thank you for this helpful suggestion. We have included additional examples and supporting references. These were integrated into the new paragraph 4.4, which discusses epigenetic memory and related miRNA regulation.
Reviewer’s Comment 7: Section 5.3 – “Integrating Machine Learning and Multi-Omics”: The information presented is brief. Please expand this section.
Authors’ Response: We fully agree. Section 5.3 at line 718 has been significantly expanded to include additional studies, providing a deeper explanation of how epigenomic and multi-omics data, when integrated with machine learning approaches, can help identify biomarkers and regulatory networks in chronic kidney disease.
Final Remarks
We sincerely thank reviewers for their thoughtful and detailed feedback. The revisions have notably improved the manuscript’s clarity, structure, and scientific rigor. We believe the current version addresses all concerns raised and is now suitable for publication in Genes.